# Targeted DNA Methylation Using Modified DNA Probes: A Potential Therapeutic Tool for Depression and Stress-Related Disorders

**DOI:** 10.3390/ijms26125643

**Published:** 2025-06-12

**Authors:** Nishtaa Modi, Jeffrey Guo, Ryan A. Lee, Alisha Greenstein, Richard S. Lee

**Affiliations:** Department of Psychiatry and Behavioral Sciences, Johns Hopkins University School of Medicine, 720 Rutland Avenue, Ross 1068, Baltimore, MD 21205, USA; nmodi4@jhu.edu (N.M.); jguo65@jhu.edu (J.G.); ral021@ucsd.edu (R.A.L.); agreenstein@colgate.edu (A.G.)

**Keywords:** epigenetics, DNA methylation (DNAm), gene expression, cortisol, FK506 binding protein 5 (FKBP5), glucocorticoid response element (GRE), monoamine oxidase A (MAOA)

## Abstract

Epigenetic modifications play a crucial role in gene regulation and have been implicated in various physiological processes and disease conditions. DNA methylation (DNAm) has been implicated in the etiology and progression of many stress-related psychiatric behaviors, such as depression. The ability to manipulate DNAm may provide a means to reverse and treat such disorders. Although CRISPR-based technologies have enabled locus-specific DNAm editing, their clinical applicability may be limited due to immunogenicity concerns and off-target effects. In this study, we introduce a novel approach for targeted DNAm manipulation using single-stranded methylated DNA probes. The probes were designed against the GRE of FKBP5 and the promoter region of MAOA. In both human embryonic kidney HEK293 and mouse pituitary AtT-20 cells, transfection with their respective methylated probes significantly increased DNAm at targeted CpG sites in a persistent and dose-dependent manner. Importantly, the induced methylation effectively attenuated glucocorticoid-induced upregulation of FKBP5 gene expression. Alteration of methylation was specific to single-stranded probes, as double-stranded methylated probes and unmethylated probes showed no significant effects. Some limitations include the need to further characterize factors that influence probe efficiency, such as probe length and CpG density; develop an efficient in vivo probe delivery system; and perform a more extensive consideration of possible off-target effects. Despite these limitations, our findings suggest that methylated DNA probes have the potential to function as a simple tool for targeted epigenetic manipulation and serve as a safer alternative to CRISPR-based epigenome editing tools for the treatment of stress-related disorders such as depression.

## 1. Introduction

Epigenetic modifications, such as DNA methylation (DNAm), can lead to alterations in the nuclear architecture and landscape, which can in turn affect the accessibility of transcription and regulatory factors to genes. DNAm typically occurs at the 5′ end of cytosine residues within CpG dinucleotides and is generally associated with transcriptional repression, which is mediated by the binding of methyl-CpG binding proteins and recruitment of histone-modifying enzymes [1]. Aberrant DNAm has been linked to many diseases, from cancer to neurodegenerative disorders, but also psychiatric diseases such as depression, anxiety, and PTSD [2,3,4,5,6].

Epigenetic mechanisms can also mediate the impact of adverse environmental conditions on gene function. For instance, environmental stressors can cause epigenetic changes in the brain [7,8]. In such instances, it has been shown that the glucocorticoid (GC) receptor that binds to cortisol can directly alter DNAm of genes that are targets of GC signaling, thus raising the possibility of mitigating disease symptoms by potentially reversing these epigenetic marks.

Targeted manipulation of DNAm at specific gene loci can modulate gene function and provide new therapeutic strategies for disorders associated with aberrant epigenetic regulation. In plants, small RNAs have been shown to direct DNAm and gene silencing [9]. Studies have shown the possibility of using pharmacological methods, such as DNMT inhibitors, to impact DNAm in mammalian systems. However, these drugs target the epigenetic machinery and are not locus-specific [10]. Although recent advances in CRISPR-based technologies have enabled locus-specific DNAm or epigenome editing in mammalian systems [11,12,13], these methods may have limited clinical applicability due to their immunogenicity [14,15,16] and off-targeting [17,18]. However, CRISPR-based therapies are now on the horizon for the treatment of debilitating diseases such as sickle cell disease [19], cancer [20], and, most recently, a rare condition called carbamoyl phosphate synthase 1 (CPS1) deficiency [21]. These therapies must weigh the benefits of genetic manipulation against potential disease burden caused by the therapy itself. Until these technologies are further refined to minimize their own disease burden and gain traction for the treatment of more common disorders, safer methods need to be explored.

In this study, we sought to develop a simpler tool for site-specifically altering DNAm. We asked whether a single-stranded methylated DNA probe can induce DNAm of its complementary target in the cell. The benefits of such a technology are that a longer probe (>20 nucleotides) would reduce off-target effects, while its considerably smaller DNA size compared to a viral vector would enable its encapsulation in naturally occurring lipid particles such as extracellular vesicles that have low immunogenicity and can cross the blood–brain barrier [22]. We tested the ability of a simple DNA probe to add DNAm onto its target genomic DNA by investigating the action of glucocorticoids on the epigenome.

Previously, we have shown that of the thousands of genomic regions in blood and brain tissues that undergo DNAm changes in response to chronic GC exposure, more than 70% are loss of DNAm events [23]. Therefore, the ability of the glucocorticoid receptor (GR) to promote the loss of DNAm of its targets provides a useful system for testing alternative epigenetic tools that can restore DNAm. Also, a successful implementation of such a tool can serve as a proof-of-concept demonstration of a novel therapeutic tool for the treatment of depression and stress-related disorders.

We focused on the FK506 binding protein 5 (FKBP5) gene, which encodes a co-chaperone of the glucocorticoid receptor and has been identified as a key regulator of the stress response [24]. It is thought that GC-induced loss of DNAm and increase in *FKBP5* levels lead to attenuated intracellular GC signaling and GC resistance, which are comorbid in more than 50% of cases of depression [25]. As such, genetic and epigenetic variations in *FKBP5* have been linked to depressive symptoms [26,27]. At the molecular level, chronic exposure to stress or excess glucocorticoids can induce the persistent demethylation of intronic glucocorticoid response elements (GREs) in the FKBP5 gene [28]. This demethylation allows for increased binding of the GR to GREs and a more robust transcription of *FKBP5*, which in turn leads to decreased sensitivity to GCs and GC resistance [29,30]. Epigenetic alterations in *FKBP5* have been linked to several stress-related psychiatric disorders such as PTSD, anxiety, and alcohol abuse [31,32,33].

We tested a probe designed against the conserved intronic GRE of human and mouse *FKBP5*. Our findings suggest that methylated DNA probes may serve as a promising tool for targeted epigenetic manipulation and have potential therapeutic applications for mitigating the impact of excess stress or GC exposure in psychiatric disorders such as depression and non-psychiatric disorders associated with aberrant DNAm patterns. Further, it may be refined to target other genes in a simpler and safer way that can circumvent some of the limitations posed by CRISPR and other epigenome-editing technologies.

## 2. Results

### 2.1. The Use of Methylated, Single-Stranded Probe to Induce Target-Specific DNAm

To test whether a simple DNA probe can target-specifically alter DNAm, we generated a probe against specific CpG sites within the glucocorticoid response element (GRE) of the human *FKBP5* gene. In vitro-methylated, single- or double-stranded DNA probes were generated and transfected into human HEK293 cells. Probes were designed with sufficiently short lengths to avoid amplification by subsequent epigenetic assays (Figure 1). Two days post-transfection, cells were analyzed for DNAm levels of the *FKBP5* GRE by bisulfite pyrosequencing. Pyrosequencing analysis showed an increase of 20.5% in DNAm at CpG-1 (*p* = 7.0 × 10^−4^) and an increase of 15.0% at CpG-2 (*p* = 4.9 × 10^−4^) in 500 ng of methylated, single-stranded probe compared to untransfected samples (Figure 2A). We also observed a dose–response when we performed transfections with only 250 ng of probe, with CpG-1 showing a more modest increase of 8.6% (*p* = 0.0066) and CpG-2 showing an increase of 7.6% (*p* = 1.1 × 10^−4^) compared to untransfected samples. Samples transfected with unmethylated, single-stranded, or methylated, double-stranded probes did not lead to an appreciable increase in DNAm (*p* > 0.05). Representative pyrosequencing tracing is shown in Figure 2B.

### 2.2. The Effect of Probe-Induced DNAm on Gene Expression

We also investigated whether the increase in DNAm was associated with differential gene expression. Previous studies have shown that while DNAm changes may not immediately result in changes in gene expression, they can modulate the level to which a gene can respond to a stimulus. In this case, the GREs of *FKBP5* are responsive to glucocorticoids in a methylation-sensitive way [29]. Treatment of the 293HEK cells with dexamethasone (DEX) caused a significant increase in FKBP5 expression (2.5-fold, *p* = 0.036). While transfection with unmethylated, single-stranded or methylated, double-stranded probes did not lead to an appreciable attenuation of the DEX-induced increase in expression, methylated, single-stranded probes significantly reduced the DEX-induced increase in *FKBP5*. Specifically, cells treated with methylated, single-stranded 500 ng of probe DNA showed a 76.2% reduction in expression compared to that of unmethylated, single-stranded 500 ng of probe DNA treated with DEX. Furthermore, we observed a dose–response with methylated, single-stranded 250 ng of probe DNA, where a 29.2% reduction in expression was observed compared to unmethylated, single-stranded 500 ng of probe DNA treated with DEX (Figure 2C).

### 2.3. Persistence of Probe-Induced DNAm Patterns and Accumulation of DNAm Following Multiple Probe Transfections

Previously, we demonstrated that glucocorticoid-induced DNA methylation persisted throughout development [28]. This persistence was replicated in a simpler mouse neuronal cell line treated with DEX [29]. Therefore, we tested whether DNA probe-induced increase in DNA methylation can persist through several weeks. Cells transfected once with 500 ng of methylated DNA probe showed an appreciable increase in DNAm across all 5 CpGs compared to unmethylated DNA probe (*p* < 0.038) at Week 1 (W1). Further culturing the transfected cells for four weeks showed that those transfected once with a methylated DNA probe retained their DNA methylation patterns at CpG-1, CpG-3, and CpG-5 (*p* < 0.0054) compared to cells transfected with an unmethylated DNA probe at Week 1 or Week 4 (Figure 3A). We observed no significant increase in DNAm at CpG-2 at Week 1 and hence no significant increase at Week 4. At CpG-4, we observed a significant increase in DNAm by 10.4%, but this probe-induced increase in DNAm showed a decay of 5.2% by Week 4 (*p* = 0.003). We also tested whether repeated transfections could have a cumulative effect on DNA methylation. 293HEK cells were transfected once, twice, three times, or four times with the same amount of DNA probes, each transfection taking place three days following the previous transfection. Results showed a cumulative effect on CpGs 3, 4, and 5, with a more profound increase in DNAm at CpGs 1 and 2 following the third and fourth transfections (Figure 3B).

### 2.4. Effect of DNA Probes in Non-Targeted Regions

We also tested the effect of our targeted probe on non-targeted regions. Chromosome conformation capture experiments have demonstrated physical interactions between GREs and the promoter [29,34], and it has been speculated that GREs that are scattered across several intronic regions of a glucocorticoid-responsive gene interact in the 3-D space of the nucleus to coordinate glucocorticoid-induced gene expression. It may be possible that other genomic regions in proximity with the region targeted by the probe may be exposed to the DNA methylation machinery. Therefore, we assayed the DNAm levels of a GRE immediately adjacent to the probe-targeted GRE in intron 5, a GRE in intron 7, and a GRE immediately downstream of the first exon in intron 1. DNAm analysis showed that most of the CpGs tested in introns 5 and 7 were too hypermethylated to undergo probe-induced DNA methylation (Figure 4A,B). CpG-6 in intron 7, which was strangely hypomethylated compared to other CpGs (<2% for all treatment groups), did not show any increase in DNAm. Further, hypomethylated intron 1 CpGs showed no differences among untransfected cells, cells treated with the unmethylated probe, and cells treated with the methylated probe. The only exception was at CpG-1, where cells transfected with the methylated probe showed a small 1.3% increase in DNAm compared to the other groups (*p* = 0.004, Figure 4C).

### 2.5. Epigenetic and Transcriptional Effects of Methylated, Single-Stranded Probe in a Mouse Pituitary Cell Line

We also tested mouse pituitary cells to determine whether the findings in 293HEK can be recapitulated in a different cell type and species. Mouse AtT-20 cells were transfected with the mouse version of the probe against the conserved *Fkbp5* intron 5 GRE. Results showed that the use of methylated, single-stranded probes increased DNAm at three of the four CpGs when compared to unmethylated single-stranded probes: CpG-2 (9.5%, *p* = 0.003), CpG-3 (6.1%, *p* = 0.003), and CpG-4 (7.2%, *p* = 0.03). There were no appreciable differences in DNAm between unmethylated, single-stranded probes and methylated, double-stranded probes (Figure 5A). We then tested whether transfected DNA probes can modulate DEX-inducibility as the human probes had in 293HEK cells. *Fkbp5* expression analysis showed that samples treated with no DNA probe, a methylated double-stranded probe, or an unmethylated single-stranded probe had similar levels of gene induction by DEX treatment (7.8-, 6.3-, and 6.7-fold induction, respectively, and *p* < 1.3 × 10^−4^). However, samples transfected with methylated, single-stranded probe showed a significant reduction in DEX-induced expression compared to samples transfected with unmethylated, single-stranded probe (42.5% reduction, *p* = 6.4 × 10^−5^, Figure 5B).

### 2.6. Additional Genomic Target of DNAm Probe: MAOA

To demonstrate the broader applicability of our methylated probe approach beyond the *FKBP5* gene, we tested the regulatory intronic region of Monoamine Oxidase A (*MAOA*) [35], a gene whose encoded protein metabolizes monoamine neurotransmitters such as dopamine, serotonin, and norepinephrine. Following the transfection of methylated single-stranded probes targeting the CpG-dense *MAOA* intronic region into 293HEK cells, we observed significant increases in DNAm across multiple CpG sites compared to cells transfected with unmethylated probes: CpG-1 (10.1%, *p* = 0.01), CpG-5 (12.7%, *p* = 0.04), CpG-6 (11.2%, *p* = 0.02), CpG-8 (9.3%, *p* = 0.006), CpG-10 (11.8%, *p* = 0.006), and CpG-12 (14.3%, *p* = 0.01) (Figure 6A). A one-way ANOVA revealed a significant difference between the two treatments (F(1, 26) = 6.25, *p* = 0.019), where methylation levels were higher in the methylated probe-treated group (M = 58.68, SD = 7.52) compared to the unmethylated probe-treated group (M = 52.31, SD = 10.25). Consistent with the increased methylation, we observed a corresponding decrease in MAOA gene expression. qRT-PCR analysis revealed that cells transfected with methylated probes showed a 27.3% reduction in *MAOA* expression compared to cells transfected with unmethylated probes (*p* = 0.041) (Figure 6B).

## 3. Discussion

In this study, we introduced a simple approach for inducing targeted DNAm using methylated DNA probes. Our approach consisted of PCR amplification against a region of interest, which in this case was a methylation-sensitive, glucocorticoid response element in the intronic region of the *FKBP5* gene [29]. The PCR product was first in vitro methylated using the bacterial methyltransferase M.SssI and denatured to yield single-stranded DNA probes. Using DNA probes against the *FKBP5* GRE, we observed appreciable increases in DNAm across at least two of the five CpGs across the GRE in the human 293HEK cells. The observed increase in DNAm was specific to the single-stranded methylated probe, as the double-stranded methylated probe and single-stranded unmethylated probe did not induce an increase in DNAm. Importantly, the increase in DNAm at these CpGs was able to attenuate the DEX-induced increase in *FKBP5* levels. Further, gene regulation by the DNA probe was dose-dependent, as the introduction of half the amount of DNA probe resulted in a reduced addition of DNAm and reduced attenuation of *FKBP5* induction by DEX compared to the full amount of DNA probe.

We also observed the persistent effects of our DNA probes on genomic target methylation, as a single administration of the DNA probe lasted through at least a four-week period, suggesting robustness of the probe-induced DNA methylation. This phenomenon is reminiscent of glucocorticoid-induced loss of DNAm, which persisted for at least four weeks in the mouse brain and in a DEX-treated neuronal cell line [28,29]. It should be noted that the reversal of DNAm induction observed at CpG-4 in Figure 3A is not new, as some of the CpGs that lost DNAm by glucocorticoid treatment also showed a similar reversion to baseline by the fourth week [29]. In addition, repeated introduction of the probe had a cumulative effect on DNA methylation, although the increase in DNAm with successive transfections was not at the same magnitude as that following the first transfection.

Investigation of other GREs in intronic regions that have been thought to physically interact in 3D space with the intronic GRE (intron 5) targeted by the probe did not show any significant changes in DNAm, emphasizing the specificity of probe targeting. One exception to the negative results at other GREs is the increase in DNAm at one CpG in intron 1 (Figure 4C). However, the magnitude of the difference was less than 2%, which is less than the 2.98% mean coefficient of variation obtained from the FKBP5 Intron 1 pyrosequencing assay.

We then tested whether the probe-induced DNAm was unique to human cells by testing the mouse version of the *Fkbp5* probe in the AtT-20 mouse pituitary cell line. Although more subtle than that observed in the human cell lines, the increase in DNAm was significant. Similarly, this increase in DNAm was able to thwart DEX-induction of *Fkbp5*. There are few factors that can potentially explain the difference in the magnitude of the probe-induced increase in DNAm, including cell-type and species-specific differences, growth rate differences in cells, and different levels of DNAm-modifying enzymes. Growth rate may play an important role, as some methylation-altering events, such as glucocorticoid-induced loss of methylation, depend on cell proliferation [29].

An additional genomic region was tested to further generalize the probe-induced increase in DNAm. This time we designed a probe against an important regulatory region of *MAOA*, whose methylation levels correlated with enzyme levels determined by PET brain imaging [35]. In addition, the MAOA enzyme metabolizes the neurotransmitter serotonin and is thus targeted by a class of antidepressants called monoamine inhibitors (MAOI). Although significant increases in DNAm were observed across many CpGs, many CpGs were impervious to the probe. The sub-optimal increase in DNAm at these CpGs may be due to the sequence of the regulatory region in *MAOA*. It is also possible that CpG-dense regions may be harder to modify, as the *MAOA* probe covers approximately the same base pairs of DNA as that for *FKBP5* but contains almost three times more CpGs. Additional work is needed to further examine the relationship between probe CpG density and DNAm efficiency.

We posit that of the three epigenetic mechanisms, i.e., DNA methylation, histone modifications, and RNA-mediated silencing, DNA methylation is the most stable, as methyltransferases faithfully copy methylation patterns during cell replication across development and cell lineage maintenance [36]. In contrast, histone modifications lack locus specificity or basepair resolution and are prone to off-target effects such as cell toxicity arising from broad effects of histone modifications over multiple loci [37]. Importantly, depending on the chromatin state, there can be rapid turnover of histone modifications on timescales of minutes to hours [38,39]. Likewise, RNA-mediated silencing involving shRNAs, siRNAs, or microRNAs is also of finite duration, as these silencing RNAs have a half-life of 28–220 h, and their continued presence is required for gene silencing [40,41]. RNA-mediated silencing can also be prone to immune activation due to the double-strandedness of silencing RNAs and off-target effects due to their relatively short nucleotide length [42,43]. Our results suggest that the DNAm-based tool may be a viable therapeutic tool given the persistence of induced DNAm and specific genomic targeting due to its longer nucleotide length.

Our current study highlights the potential of this method to modulate stress response pathways, as demonstrated by the targeted methylation of the *FKBP5* gene. *FKBP5* is known to be influenced by glucocorticoid exposure and methylation status, with demethylation of the glucocorticoid response element leading to increased gene expression upon subsequent glucocorticoid exposure [29]. By inducing targeted methylation of this region, we were able to attenuate the glucocorticoid-induced upregulation of FKBP5 expression, suggesting that our approach could be used to fine-tune stress response pathways and potentially mitigate the detrimental effects of chronic stress or glucocorticoid exposure.

The success of our method in both human and mouse cell lines indicates its potential for translation to in vivo models and eventual clinical applications. However, several challenges need to be addressed before this approach can be fully realized in vivo. First, the efficiency of probe delivery and cellular uptake may vary depending on the cell type and target tissue, requiring optimization of tissue uptake methods and probe design. Second, the long-term stability and persistence of the induced methylation changes need to be evaluated in vivo to determine the durability of the therapeutic effects. The safety and potential off-target effects of the methylated probes must also be thoroughly assessed in animal models before considering human applications. Finally, we will need a suitable delivery device that can tissue-specifically deliver DNA probes for in vivo studies.

Small extracellular vesicles (EVs) such as exosomes hold immense therapeutic potential as delivery devices, as their vesicle cargo space can easily accommodate a ~200 bp DNA fragment and their cell and tissue uptake efficiency can be modified [44]. Recent advances in the EV field have provided promising directions on how surface proteins of EVs can be engineered to increase tissue and cell uptake efficiency by overexpressing certain surface proteins obtained from donor cells on the surface of EVs, which can more efficiently target similar tissue types [45,46,47,48]. This “like-attracts-like” phenomenon is thought to be mediated by core ligands and homing peptides that fuse with similar transmembrane proteins on the EV surface to confer affinity to target cells [49,50]. Alternatively, click chemistry has enabled the conjugation of peptides of interest on the surface of EVs to increase their delivery to target tissues [51,52]. Such modifications will enable the efficient delivery of small molecules, such as our DNA probes, to brain tissues.

There are several limitations to the study. We have not tested several important factors that may influence the efficacy of the DNA probe. These include CpG density, length of the probe, and cell growth rate. First, it is unclear whether a highly CpG-dense region is less likely to become methylated compared to a genomic region with lower CpG density. The *MAOA* region, which harbors 14 CpGs over ~170 bps, was significantly more CpG-dense than the *FKBP5* GRE with 5 CpGs over ~200 bps of DNA. Second, it is unclear whether the length of the probe can affect targeting and methylation efficiency. While longer DNA probes can provide better target specificity compared to 20 nucleotide-long guide RNAs used in CRISPR, longer sequences can also contain small regions of high complementarity that can hinder efficient targeting. As such, additional non-targeted regions need to be examined in greater detail. Third, it remains unclear whether cell proliferation rate can affect target DNA methylation. We have only used two cell lines that undergo robust cell division in culture. Additional cell lines, especially those of neuronal origins, need to be tested to be useful in vivo for animal models of stress and depression.

Despite these challenges, our study provides a proof-of-concept demonstration for the use of methylated DNA probes as a targeted epigenetic therapy. The ability to induce site-specific DNAm changes can open new avenues for the treatment of a wide range of diseases characterized by aberrant methylation patterns. As our understanding of the epigenetic basis of diseases continues to grow, the development of targeted epigenetic therapies, such as the one presented here, will become increasingly important. In conclusion, we have developed a novel method for targeted induction of DNAm using methylated DNA probes. This approach offers a promising tool for modulating gene expression and function, with potential therapeutic applications in various diseases characterized by aberrant methylation patterns.

## 4. Methods

### 4.1. Probe Design and Amplification

Methylated DNA probes targeting the conserved glucocorticoid response element (GRE) in intron 5 of the human (chr6: 35,601,961–35,602,194; GRCh38/hg38, 256 bp) and mouse (chr17: 28,639,321–28,639,560; GRCm39/mm39, 239 bp) of *FKBP5* were designed. A separate DNA probe designed against an intronic, regulatory region of the human *MAOA* gene (chrX: 43,656,383–43,656,553; GRCh38/hg38, 170 bp) was also tested [35]. Additional tests for non-specific effects of the human *FKBP5* probe investigated another adjacent region in intron 5 (chr6: 35,610,962–35,611,313), intron 1 (chr6: 35,687,767–35,688,045), and intron 7 (chr6: 35,590,524–35,591,014). For both human and mouse probes, PCR primers targeted smaller regions than those analyzed by bisulfite sequencing to preclude the amplification of probe DNA during methylation analysis. Primers used for generating the probes are shown in Table 1. The genomic organization of the human *FKBP5* locus is shown in Figure 1.

### 4.2. In Vitro Methylation of DNA Probes

Purified *FKBP5* probes were subjected to in vitro methylation using the bacterial CpG methyltransferase (M.SssI, New England Biolabs, Ipswich, MA, USA) [53]. One µg of the probe was incubated with 4 units of M.SssI, 160 µM S-adenosylmethionine (SAM), and 1X NEBuffer 2 in a total reaction volume of 100 µL at 37 °C for two 1 h cycles followed by 20 min at 65 °C for enzyme inactivation. A negative control reaction (*FKBP5* SssI-) was performed in parallel, where the M.SssI enzyme was replaced with an equal volume of water. Following in vitro methylation, probes were purified again and eluted in 20 µL of TE buffer. The concentrations of the methylated (FK SssI+) and unmethylated (FK SssI-) probes were measured using Qubit 4 (Thermo Fisher Scientific, Waltham, MA, USA). Similar reactions were performed for the mouse *Fkbp5* and human *MAOA* probes.

### 4.3. Cell Culture and Transfection

To induce demethylation at the GRE of the endogenous *FKBP5* gene, we chose cell lines that we have previously shown to undergo DEX or glucocorticoid-induced loss of methylation [29,54]. Human embryonic kidney 293 (293HEK) and mouse pituitary AtT-20 cells were purchased from Atcc.org and were treated with 1 µM dexamethasone (DEX) for 5 days and cultured for an additional 5 days without any DEX. This would allow CpGs to undergo persistent loss of DNA methylation, which would then be restored by introducing methylated probes. Separate wells of cells were left untreated as negative controls or transfection reagent-only controls. Prior to transfection, cells were seeded in 24-well plates at a density of 2 × 10^5^ cells per well and allowed to adhere overnight in DMEM free of antibiotics. The transfection of the methylated and unmethylated probes was performed in quadruplicate (500 ng of probe per well) using X-tremeGENE 360 Transfection Reagent (MilliporeSigma, Burlington, MA, USA) according to the manufacturer’s instructions. Cells were fed fresh DMEM media one day after transfection and harvested on the second day for collecting gDNA. For the assessment of glucocorticoid-induced gene expression, a subset of the transfected 293HEK cells was treated with 1 µM DEX for 4 h prior to harvesting. Cells were harvested for total RNA extraction using the RNeasy Mini Kit (Qiagen, Germantown, MD, USA) according to the manufacturer’s instructions.

### 4.4. DNA Extraction and Methylation Analysis by Bisulfite Pyrosequencing

Genomic DNA was extracted from 293HEK and AtT-20 cells using the DNeasy Blood and Tissue Kit (Qiagen) according to the manufacturer’s instructions. The extracted DNA was bisulfite-converted using the EZ DNAm-Gold Kit (Zymo Research, Irvine, CA, USA) according to the manufacturer’s instructions. For bisulfite pyrosequencing, bisulfite-converted DNA and a pair of outside primers were used to PCR amplify the genomic region of interest. Two μLs of outside PCR products were used for a second PCR reaction using inside PCR primers. One of the Inside primers was biotinylated and allowed for the isolation of and primer extension on a PSQ HS 96 pyrosequencer (Qiagen) according to the manufacturer’s instructions, and CpG sites were quantified, from 0% to 100% methylation, using the Pyro Q-CpG software. Primers used for assessing DNAm levels of *FKBP5* and *MAOA* genes are shown in Table 1. We determined the coefficient of variation (COV) using pyrosequencing assays for the *MAOA*, *FKBP5* Intron 1, and Intron 5 loci. This was performed by running six replicates of bisulfite PCR products on the same pyrosequencing plate and measuring the variation in %DNAm for multiple CpGs. From these assays, we determined the COV to be 3.66% for *MAOA*, 2.98% for *FKBP5* Intron 1 GRE, and 3.27% for FKBP5 Intron 5 GRE.

### 4.5. Gene Expression Analysis

Reverse Transcription Quantitative Real-Time PCR (RT-qPCR) was performed to assess the impact of targeted DNAm on gene expression. Total RNA from 293HEK or AtT-20 RNA samples was extracted using the RNeasy Mini Kit, and complementary DNA (cDNA) was synthesized using the QuantiTect Reverse Transcription Kit (Qiagen) according to the manufacturer’s instructions. *FKBP5* and *MAOA* expression analysis was performed using Taqman probes and the QuantStudio5 Real-Time PCR System (Thermo Fisher Scientific). For the determination of relative expression values, the −ΔΔCt method was used, where triplicate Ct values for each cDNA sample were averaged and subtracted from those derived from *ACTB* [55].

### 4.6. Statistics and Data Analysis

Data are presented as mean ± SEM, with the number of replicates (N) indicated in the figure legends. Prior to hypothesis testing, data distributions were assessed for normality using the Shapiro–Wilk test. For comparisons between two groups (e.g., methylated vs. unmethylated probe or DEX-treated vs. vehicle control), unpaired two-tailed Student’s *t*-tests were performed. For experiments involving multiple CpG sites, e.g., the *MAOA* locus, one-way ANOVA was used to test for overall differences, followed by Tukey’s post hoc test for pairwise comparisons. All analyses were conducted in GraphPad Prism v.9. A *p*-value < 0.05 was considered statistically significant.

## Figures and Tables

**Figure 1 ijms-26-05643-f001:**
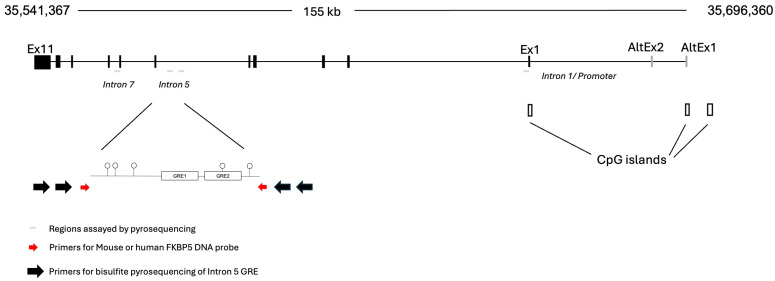
Genomic organization of *FKBP5*. The human FKBP5 locus is located on the negative strand of chromosome 6. For this study, three intronic regions indicated by thin horizontal gray lines have been tested. Two sets of large black arrows represent the outside and inside PCR primers used for bisulfite pyrosequencing. These black arrows flank the primers used for generating the probe (red arrows), which cannot be amplified by the pyrosequencing primers. The intron 5 GREs are composed of the main GRE formed by GRE1 and GRE2, for which the DNA probe was designed, and an adjacent GRE that was tested as a negative control region.

**Figure 2 ijms-26-05643-f002:**
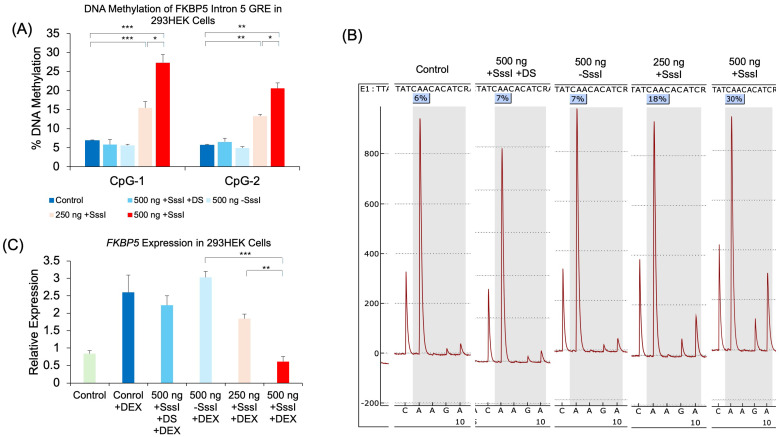
Dose-dependent DNAm and gene expression changes following transfection of methylated DNA probes against *FKBP5*. (**A**) 293HEK cells were transfected with a single-stranded unmethylated probe (500 ng − SssI), a single-stranded methylated probe at two different concentrations (250 ng and 500 ng + SssI), or a double-stranded methylated probe (500 ng + SssI +DS). Untransfected cells served as controls (Control). DNAm levels of five CpG sites at the conserved glucocorticoid response element (GRE) of human *FKBP5* intron 5 were analyzed by bisulfite pyrosequencing. Data for the first two CpGs are shown. (**B**) Typical pyrograms obtained from bisulfite pyrosequencing are shown for each group at CpG-1. The percent DNAm determination occurs when R (or A/G) is dispensed, and it corresponds to the reverse complement of T/C (T for unmethylated and C for methylation CpG, respectively). Each pyrogram represents % methylation from one sample. (**C**) *FKBP5* expression was measured by qRT-PCR in the same groups of 293HEK cells as in (**A**) treated with 1 μM dexamethasone (DEX) for four hours prior to collection. Bar graphs represent the mean ± SEM, and unpaired two-tailed Student’s *t*-tests were performed with *n* = 4 per group. *** *p* < 0.001, ** *p* < 0.01, and * *p* < 0.05.

**Figure 3 ijms-26-05643-f003:**
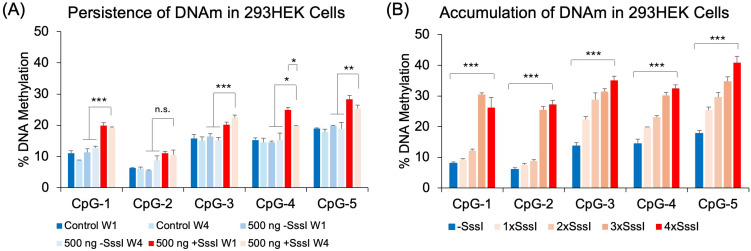
Persistence and accumulation of DNA methylation. (**A**) 293HEK cells were transfected with the unmethylated (−SssI) and methylated (+SssI) *FKBP5* DNA probe at Week 1 (W1), after which the cells were cultured for an additional four weeks (W4) before analysis by bisulfite pyrosequencing. (**B**) 293HEK cells were transfected with unmethylated (−SssI) and methylated (+SssI) *FKBP5* DNA probes consecutively every three days and expanded, while 50% of the cells were collected for analysis. Bar graphs represent the mean ± SEM, and unpaired two-tailed Student’s *t*-tests were performed with *n* = 4 per group. *** *p* < 0.001, ** *p* < 0.01, and * *p* < 0.05.

**Figure 4 ijms-26-05643-f004:**
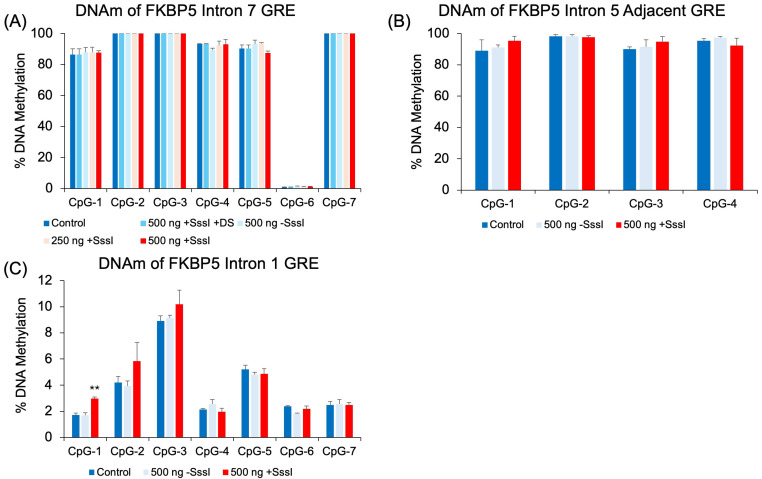
DNAm analysis of human *FKBP5* GREs at additional intronic regions. Experimentally verified glucocorticoid response elements (GREs) at intron 7 (**A**), intron 5 (**B**), and 1 (**C**) were evaluated in probe-transfected 293 HEK samples for non-specific epigenetic effects. There were no significant differences in cells transfected with unmethylated vs. methylated DNA probes, except at CpG-1 of Intron 1 (**C**). Bar graphs represent the mean ± SEM, and unpaired two-tailed Student’s *t*-tests were performed with *n* = 4 per group. ** *p* < 0.01.

**Figure 5 ijms-26-05643-f005:**
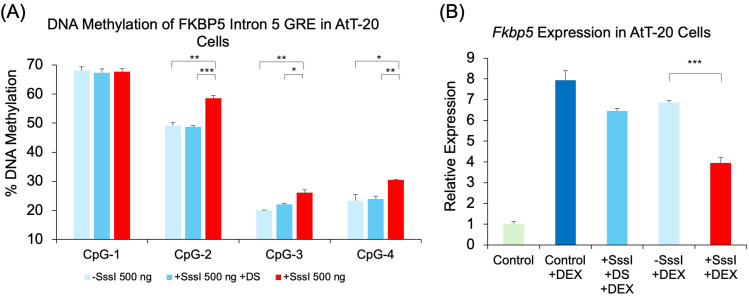
DNAm and gene expression analysis following transfection of methylated DNA probes against mouse *Fkbp5*. (**A**) AtT-20 cells were transfected with single-stranded unmethylated probe (500 ng − SssI), single-stranded methylated probe (500 ng + SssI), or double-stranded methylated probe (500 ng + SssI +DS). Untransfected cells served as controls (Control). DNAm levels of four CpG sites at the conserved glucocorticoid response element (GRE) of mouse *Fkbp5* intron 5 were analyzed by bisulfite pyrosequencing. Data for all four GRE CpGs are shown. (**B**) *Fkbp5* expression was measured by qRT-PCR in the same groups of AtT-20 cells as in (**A**) treated with 1 μM dexamethasone (DEX) for four hours prior to collection. Bar graphs represent the mean ± SEM, and unpaired two-tailed Student’s *t*-tests were performed with *n* = 4 per group. *** *p* < 0.001, ** *p* < 0.01, and * *p* < 0.05.

**Figure 6 ijms-26-05643-f006:**
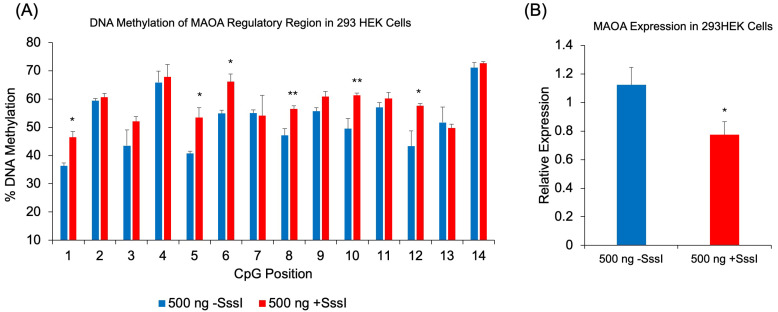
DNAm and gene expression analysis following transfection of methylated DNA probes against *MAOA*. (**A**) 293HEK cells were transfected with a single-stranded unmethylated probe (-SssI) or a single-stranded methylated probe (+SssI). DNAm levels of 14 CpG sites at a regulatory region of human *MAOA* were analyzed by bisulfite pyrosequencing. (**B**) *MAOA* expression was measured by qRT-PCR in the same groups of 293HEK cells. Bar graphs represent the mean ± SEM, and unpaired two-tailed Student’s *t*-tests were performed with *n* = 4 per group. * *p* < 0.05 and ** *p* < 0.01. A one-way ANOVA was used to test for overall differences across all 14 CpGs, followed by Tukey’s post hoc test for pairwise comparisons.

**Table 1 ijms-26-05643-t001:** Sequence of primers for probe design and pyrosequencing.

Probe Primers	Sequence	Size
Human FKBP5 Probe-Forward	5′-AAAGTCAAACCAAACCAAATTACC -3′	256 bp
Human FKBP5 Probe-Reverse	5′-TTTGTTACTGCTGTGCACTCTCT -3′	
Human MAOA Probe-Forward	5′-TCGACGTAGTCGTGATCGG -3′	170 bp
Human MAOA Probe-Reverse	5′-GCAGGATATGGGGCCAAG -3′	
Mouse Fkbp5 Probe-Forward	5′-CAGACACCAGCTACTATAATTAG -3′	239 bp
Mouse Fkbp5 Probe-Reverse	5′-GCACATGAACTCGATGTGCTGACA -3′	
**Pyrosequencing Primers**	**Sequence**	**Size**
Human FKBP5 Intron 5 Outside-A	GGTAGAGAAAGAAATAAATAAGTTA	286 bp
Human FKBP5 Intron 5 Outside-B	TTCTTACATTTCATTTTTATTACTACTA	
Human FKBP5 Intron 5 Inside-A *	AAGATTATGTAATTTAAAGGGGGAGGG	
Human FKBP5 Intron 5 Inside-B	CTCTCTTTCCTTTTTTCCCCCCTAT	
Human FKBP5 Intron 5 Pyro 1	TCTTTCCTTTTTTCCCCCCTATT	
Human FKBP5 Intron 5 Pyro 2	CAATTTAAATAATATTTTACAACT	
Human FKBP5 Intron 5_2 Out-A	ATTTAATTGGTTTGGGTGTTAGAA	406 bp
Human FKBP5 Intron 5_2 Out-B	CCTCTCAATACTTTCAACCACA	
Human FKBP5 Intron 5_2 In-A *	GAGAATTATTGTATTGGAGGTT	
Human FKBP5 Intron 5_2 In-B	ATTCTACAAATTCCAATTATTAAC	
Human FKBP5 Intron 5_2 Pyro 1	GTATTGGAGGTTTATTGGTT	
Human FKBP5 Intron 5_2 Pyro 2	TAGATGATTATGAGTTTGGAGTT	
Human FKBP5 Intron 5_2 Pyro 3	GTTTAAGTTTTTTTTATATTTGTT	
Human FKBP5 Intron 5_2 Pyro 4	GATTTGGAGAGGGAAAGGAGGT	
Human FKBP5 Intron 1 Out-A	AGTTTAAATTGTTTTATGTAGAATTTATTGA	350 bp
Human FKBP5 Intron 1 Out-B	TCACTCCCAAACCATACC	
Human FKBP5 Intron 1 Inside-A	GTTTTGAATTATATTGAAGGGTATTT	
Human FKBP5 Intron 1 Inside-B *	CAAAACTCCTTATACTCTTCTATTCTAA	
Human FKBP5 Intron 1 Pyro 1	GTAGAATTYGATTTTAGAGA	
Human FKBP5 Intron 7 Outside-A	AGAGTGAAATTGAGATGGAAATATGT	503 bp
Human FKBP5 Intron 7 Outside-B	AATTTCTTCTCCATCCACTTCCTATA	
Human FKBP5 Intron 7 Inside-A	AGGAGGTATGTTGTTTTTGGAATTTAAG	
Human FKBP5 Intron 7 Inside-B *	AATTTATCTCTTACCTCCAACACT	
Human FKBP5 Intron 7 Pyro 1	GGAGAAGTATAAAAAAAAAATGG	
Human FKBP5 Intron 7 Pyro 2	GTTATAGAGTTTAGTGGTTT	
Human FKBP5 Intron 7 Pyro 3	GGAGTTATAGTGTAGGTTTT	
Human FKBP5 Intron 7 Pyro 4	TTAAGGAGTTATTTGGTAGA	
Human FKBP5 Intron 7 Pyro 5	TGATATATAGGAATAAAATAAGAAT	
Human MAOA Outside-A	GATTTAGGAGYGTGTTAGTTAAAGT	278 bp
Human MAOA Outside-B	TTATTATATCTACCTCCCCCAATC	
Human MAOA Inside-A	AGTTAAAGTATGGAGAATTAAG	
Human MAOA Inside-B *	ATCTACCTCCCCCAATCACACCACCAAC	
Human MAOA Pyro 1	AAAGTATGGAGAATTAAGAGAAGG	
Human MAOA Pyro 2	GAGTATYGYGGGTTATATG	
Human MAOA Pyro 3	AGGTGGTATTTTAGGTTAGTGTGGA	
Mouse Fkbp5 Intron 5 Outside-A	GATGATTAGTTTTTTTTAGTAGTGATGT	308 bp
Mouse Fkbp5 Intron 5 Outside-B	CTTATTATTCTCTTACTACCCTAA	
Mouse Fkbp5 Intron 5 Inside-A	TAGTTTTTGGGGAAGAGTGTAGAGTTAT	
Mouse Fkbp5 Intron 5 Inside-B *	ATTTTAAAAAACACAAAACACCCTATT	
Mouse Fkbp5 Intron 5 Pyro 1	AGAAAAGGGAAAGTAGG	
Mouse Fkbp5 Intron 5 Pyro 2	TAGTTTTTGTTATTGTTGTATG	

* These primers have been biotinylated and HPLC-purified for pyrosequencing.

## Data Availability

The raw data supporting the conclusions of this article will be made available by the authors upon request.

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
