# Peer review of "Targeted DNA Methylation Using Modified DNA Probes: A Potential Therapeutic Tool for Depression and Stress-Related Disorders"

_ijms, 2025, doi:10.3390/ijms26125643_

Round 1

Reviewer 1 Report

Comments and Suggestions for Authors

In the present manuscript Modi et al have developed methylated DNA probes as a tool for locus specific DNA methylation for modifying gene expression in a targeted manner. The method they have developed is an important addition to gene manipulation arsenal and I look forward to demonstration of its feasibility in vivo. The manuscript is clearly written, the data support the inferences and the graphs clearly communicate the observations. I recommend the manuscript for publication. However, there are few minor points I would like the authors to address –

  • Methods section – The authors should include a “Statistics” section in the methods where the n’s, replicates, statistical tests and the software used to conduct these tests are clearly described.
  • Statistical test – Please indicate the statistical test used in the legends to Fig 2, Fig 3, Fig 4, Fig 5 and Fig. 6. ANOVA is mentioned in the results 2.6, but this should also be listed in the legend.
  • Result section 2.4 – The authors find a 1.3% increase in methylation of CpG-1 when testing effect of methylated probe on non-targeted regions. In the discussion they state- “However, the magnitude of the difference was less than 2%, which is unlikely to be functionally relevant”. The authors should either provide supporting references showing that this change is indeed functionally irrelevant, or they should provide gene expression data to confirm the same.

Author Response

In the present manuscript Modi et al have developed methylated DNA probes as a tool for locus specific DNA methylation for modifying gene expression in a targeted manner. The method they have developed is an important addition to gene manipulation arsenal and I look forward to demonstration of its feasibility in vivo. The manuscript is clearly written, the data support the inferences and the graphs clearly communicate the observations. I recommend the manuscript for publication. However, there are few minor points I would like the authors to address –

  • Methods section – The authors should include a “Statistics” section in the methods where the n’s, replicates, statistical tests and the software used to conduct these tests are clearly described.

We have now included a small section in the Methods for statistics.

  • Statistical test – Please indicate the statistical test used in the legends to Fig 2, Fig 3, Fig 4, Fig 5 and Fig. 6. ANOVA is mentioned in the results 2.6, but this should also be listed in the legend.

Statistical tests used for all figures are now included in the legends.

  • Result section 2.4 – The authors find a 1.3% increase in methylation of CpG-1 when testing effect of methylated probe on non-targeted regions. In the discussion they state- “However, the magnitude of the difference was less than 2%, which is unlikely to be functionally relevant”. The authors should either provide supporting references showing that this change is indeed functionally irrelevant, or they should provide gene expression data to confirm the same.

We have added the following to the Methods: We determined the coefficient of variation (COV) using the pyrosequencing assay for the MAOA, FKBP5 Intron 1 and Intron 5 loci. This was done by running multiple replicates of bisulfite PCR products on the same pyrosequencing plate and measuring the variation of %DNAm for multiple CpGs. From these assays, we determined the COV to be 3.66% for MAOA, 2.98% for FKBP5 Intron 1 GRE, and 3.27% for FKBP5 Intron 5 GRE, so that the 1.3% increase in DNA methylation in CpG-1 was less that the COV or within the expected measurement error.

We have also added the following to the Discussion: “However, the magnitude of the difference was less than 2%, which is less than the 2.98% mean coefficient of variation obtained from the FKBP5 Intron 1 pyrosequencing assay.”

Reviewer 2 Report

Comments and Suggestions for Authors

The paper presents a novel approach utilizing modified DNA probes for targeted DNA methylation, representing an innovative research direction with profound potential applications, particularly in the treatment of depression and stress-related disorders. The experimental design is reasonably devised, involving the investigation of multiple genes (FKBP5 and MAOA) in different cell lines (human HEK293 cells and mouse pituitary AtT-20 cells). The feasibility and effectiveness of the method are primarily validated through analyses at various levels, including DNA methylation status and gene expression. However, there are still some deficiencies in the elaboration of certain content, experimental design, and discussion analysis, which require further improvement.

Abstract Section:The abstract mentions the research method, main results, and potential applications but lacks any discussion on the limitations of the study. It is recommended to include these limitations to provide a more comprehensive overview for readers.

Some sentences can be further simplified. For instance, "We designed probes targeting the glucocorticoid response element (GRE) in the FKBP5 (FK506-binding protein 5) gene and the promoter region of the MAOA (monoamine oxidase A) gene." can be revised to "Designed probes targeting GRE in FKBP5 gene and the promoter region of MAOA gene," making the expression more concise without losing key information.

Introduction Section:The transition to the present research method seems abrupt after discussing the relationship between DNA methylation and diseases, as well as the limitations of existing techniques. Adding a few sentences to explain why the development of a new methylated DNA probe method was conceived from the shortcomings of current technologies would enhance the logical flow of the narrative.

Results Section:When discussing the effects of DNA probes on non-target regions, there is a brief mention of a small increase in DNA methylation at certain non-target CpG sites (such as CpG-1 in intron 1), stating that it is unlikely to have functional relevance. However, deeper analysis and exploration are lacking. It is suggested to further analyze the potential reasons for this change and whether such changes have been reported in other similar studies, providing more depth to the results analysis.

Discussion Section:When comparing the present method with other epigenetic mechanisms (such as histone modifications and RNA-mediated gene silencing), the stability of DNA methylation is emphasized, but there is a lack of comparative analysis on the advantages and limitations of other mechanisms in disease treatment. It is recommended to include this information to provide readers with a more comprehensive understanding of the method's position and value in the field of epigenetic therapy.

When discussing the challenges faced by the study and future perspectives, there is a dearth of specific research ideas and methods on how to address probe delivery and cell uptake efficiency issues, as well as how to evaluate the stability of methylation changes in vivo. It is suggested to propose more feasible solutions and research directions, incorporating current advancements in related fields, to enhance the study's forward-looking and operational aspects.

Conclusion Section:The conclusion section is overly simplified, focusing only on the method's innovativeness and potential applications, without summarizing the research findings. It is recommended to include a synopsis of the main results to make the conclusion more comprehensive. Additionally, mentioning the limitations of the study would render the conclusion more objective and balanced.

References:The format of references should be unified, and the majority should be from the past five years.

Author Response

The paper presents a novel approach utilizing modified DNA probes for targeted DNA methylation, representing an innovative research direction with profound potential applications, particularly in the treatment of depression and stress-related disorders. The experimental design is reasonably devised, involving the investigation of multiple genes (FKBP5 and MAOA) in different cell lines (human HEK293 cells and mouse pituitary AtT-20 cells). The feasibility and effectiveness of the method are primarily validated through analyses at various levels, including DNA methylation status and gene expression. However, there are still some deficiencies in the elaboration of certain content, experimental design, and discussion analysis, which require further improvement.

Abstract Section: The abstract mentions the research method, main results, and potential applications but lacks any discussion on the limitations of the study. It is recommended to include these limitations to provide a more comprehensive overview for readers.

We have now added the following limitations to the abstract: “Some limitations include the need to further characterize factors that influence probe efficiency, such as probe length and CpG density, develop an efficient probe delivery system, and perform a more extensive consideration of possible off-target effects.”

We have also added a more detailed limitations section to the discussion: “We have not tested several important factors that may influence the efficacy of the DNA probe. These include CpG density, length of the probe, and cell growth rate. First, it is unclear whether a highly CpG-dense region is less likely to become methylated compared to a genomic region with lower CpG density. The MAOA region, which had 14 CpGs over ~170 bps was significantly more CpG-dense than the FKBP5 GRE with 5 CpGs over ~200 bps of DNA. Second, it is unclear whether the length of the probe can affect targeting and methylation efficiency. While longer DNA probes can provide better target specificity compared to 20 nucleotide-long guide RNAs used in CRISPR, longer sequence can also provide small regions of high complementarity that can hinder efficient targeting. As such, additional non-targeted regions need to be examined in greater detail. Third, it remains unclear whether cell proliferation rate can affect target DNA methylation. We have only used two cell lines that undergo robust cell division in culture. Additional cell lines, especially those of neuronal origins, need to be tested to be useful in vivo for animal models of stress and depression.”

Some sentences can be further simplified. For instance, "We designed probes targeting the glucocorticoid response element (GRE) in the FKBP5 (FK506-binding protein 5) gene and the promoter region of the MAOA (monoamine oxidase A) gene." can be revised to "Designed probes targeting GRE in FKBP5 gene and the promoter region of MAOA gene," making the expression more concise without losing key information.

Thank you for this suggestion. We have included this edit to make the sentence more concise, “Probes were designed against the GRE of FKBP5 and the promoter region of MAOA.”

Introduction Section:The transition to the present research method seems abrupt after discussing the relationship between DNA methylation and diseases, as well as the limitations of existing techniques. Adding a few sentences to explain why the development of a new methylated DNA probe method was conceived from the shortcomings of current technologies would enhance the logical flow of the narrative.

We have restructured the introduction to emphasize some possible shortcomings of the currently available technology: “In this study, we sought to develop a simpler tool for site-specifically altering DNAm. We asked whether a single-stranded, methylated DNA probe can induce DNAm of its complementary target in the cell. The benefits of such a technology are that a longer probe (>20 nucleotides) would reduce off-target effects while its considerably smaller DNA size compared to a viral vector would enable its encapsulation in naturally occurring lipid particles such as extracellular vesicles that have low immunogenicity and can cross the blood brain barrier. We tested the ability of a simple DNA probe to add DNAm onto its target genomic DNA by investigating the action of glucocorticoids on the epigenome.”

Results Section:When discussing the effects of DNA probes on non-target regions, there is a brief mention of a small increase in DNA methylation at certain non-target CpG sites (such as CpG-1 in intron 1), stating that it is unlikely to have functional relevance. However, deeper analysis and exploration are lacking. It is suggested to further analyze the potential reasons for this change and whether such changes have been reported in other similar studies, providing more depth to the results analysis.

We have added text on the coefficient of variation calculated for pyrosequencing assays in the Methods and Discussion sections: “However, the magnitude of the difference was less than 2%, which is less than the 2.98% mean coefficient of variation obtained from the FKBP5 Intron 1 pyrosequencing assay.”

Discussion Section:When comparing the present method with other epigenetic mechanisms (such as histone modifications and RNA-mediated gene silencing), the stability of DNA methylation is emphasized, but there is a lack of comparative analysis on the advantages and limitations of other mechanisms in disease treatment. It is recommended to include this information to provide readers with a more comprehensive understanding of the method's position and value in the field of epigenetic therapy.

We have now added some text on the comparative analysis of advantages and limitations of other mechanisms in disease treatment: “We posit that of the three epigenetic mechanisms, i.e., DNA methylation, histone modifications, and RNA-mediated silencing, DNA methylation is the most stable, as methyltransferases faithfully copy methylation patterns during cell replication across development and cell lineage maintenance. In contrast, histone modifications lack locus specificity or basepair resolution and are prone to off-target effects such as cell toxicity arising from broad effects of histone modifications over multiple loci. Importantly, depending on the chromatin state, there can be rapid turnover of histone modifications on timescales of minutes to hours. Likewise, RNA-mediated silencing involving shRNAs, siRNAs, or microRNAs are also of finite duration, as these silencing RNAs have a half-life of 28-220 hours, and their continued presence is required for gene silencing. RNA-mediated silencing can also be prone to immune activation due to the double-strandedness of silencing RNAs and off-target effects due to their relatively short nucleotide length. Our results suggest that the DNAm-based tool may be a viable therapeutic tool given the persistence of induced DNAm and specific genomic targeting due to its longer nucleotide length”

When discussing the challenges faced by the study and future perspectives, there is a dearth of specific research ideas and methods on how to address probe delivery and cell uptake efficiency issues, as well as how to evaluate the stability of methylation changes in vivo. It is suggested to propose more feasible solutions and research directions, incorporating current advancements in related fields, to enhance the study's forward-looking and operational aspects.

We have now added more details on feasible solutions, research directions, and recent technologies for increasing the delivery of DNA probes to target tissues: “Small extracellular vesicles (EVs) such as exosomes hold immense therapeutic potential as delivery devices, as their vesicle cargo space can easily accommodate a ~200 bp DNA fragment and their surface proteins profiles can be modified to alter tissue targeting. Recent advances in the EV field provide tantalizing directions on how surface proteins of EVs can be engineered to increase tissue and cell uptake efficiency. Efforts have been made to engineer and modify exosomes that can target specific cells and deliver a therapeutic payload by overexpressing surface proteins obtained from donor cells on the surface of EVs, which can then target similar tissue types. This “like-attracts-like” phenomenon is thought to be mediated by core ligands and homing peptides that fuse with similar transmembrane proteins on the exosome surface to confer affinity to target cells. Alternatively, click chemistry has enabled the conjugation of peptides of interest on the surface of EVs to increase their delivery to target tissues. Such modifications will enable the efficient delivery of small molecules such as our DNA probes to brain tissues.”

Conclusion Section:The conclusion section is overly simplified, focusing only on the method's innovativeness and potential applications, without summarizing the research findings. It is recommended to include a synopsis of the main results to make the conclusion more comprehensive. Additionally, mentioning the limitations of the study would render the conclusion more objective and balanced.

References: The format of references should be unified, and the majority should be from the past five years.

We thank you for the suggestions. First five paragraphs summarize the main findings of the paper, starting with dose-dependent gene silencing, persistence and cumulative effects, non-targeted regions, mouse Fkbp5, and then human MAOA. We also added a section on the limitations of the study. Efforts have been made to include more recent references, especially when mentioning EVs. The following limitations have been added (also requested by Rev2): “There are several limitations to the study. We have not tested several important factors that may influence the efficacy of the DNA probe. These include CpG density, length of the probe, and cell growth rate. First, it is unclear whether a highly CpG-dense region is less likely to become methylated compared to a genomic region with lower CpG density. The MAOA region, which had 14 CpGs over ~170 bps was significantly more CpG-dense than the FKBP5 GRE with 5 CpGs over ~200 bps of DNA. Second, it is unclear whether the length of the probe can affect targeting and methylation efficiency. While longer DNA probes can provide better target specificity compared to 20 nucleotide-long guide RNAs used in CRISPR, longer sequence can also provide small regions of high complementarity that can hinder efficient targeting. As such, additional non-targeted regions need to be examined in greater detail. Third, it remains unclear whether cell proliferation rate can affect target DNA methylation. We have only used two cell lines that undergo robust cell division in culture. Additional cell lines, especially those of neuronal origins, need to be tested to be useful in vivo for animal models of stress and depression.”

Reviewer 3 Report

Comments and Suggestions for Authors

General remarks: Avoid statements like ‘have been published elsewhere’ with a reference. Brief description of conditions would be appreciated. 

  1. RESULTS

Human Embryonic Kidney (HEK) 293: why not neuronal or glial cells from the start.

What is the percentage of transfection.

2.2 Fig 2A compared to fig 2B: CpG-1 results in 2A compared to pyrogram 2B look confusing. Percentages are not comparable eg control 2A is 7% control 2B pyrogram is 6%.  Please check. Or leave pyrogram out because pryrogram is one analysis, fig 2A represents multiple analyses with SEM.

Mention the number of replicates.

Are replicates from different passages.

Why using SEM instead of SD?

Mention statistical test that is used.

2.3 Only the media were changed ? Thus no tranfection during the 4 weeks period took place?. Just to be sure. Can you mention the percentage of dying cells? Please clarify my questions.

2.5 ‘We also tested mouse pituitary cells to determine whether the findings in 293HEK can be recapitulated in a different cell-type and species.’ Reviewer appreciate the different cell-type, however why not using  the human neuronal or glial cells. You are interested in humans not in mice , right? Suggestion reviewer: show results in neuronal or glial cells.

  1. METHODS.

Mention X-tremeGENE 360 Transfection Reagent instead of X-tremeGENE 360

Brief description of  conditions for bisulfite pyrosequencing Is appreciated.  The remark ‘Conditions for bisulfite pyrosequencing have been published elsewhere35 ‘ is not sufficient.

Which primer design program did the authors use?

Author Response

General remarks: Avoid statements like ‘have been published elsewhere’ with a reference. Brief description of conditions would be appreciated. 

We have now added brief descriptions of experimental details for pyrosequencing: “Briefly, bisulfite-converted DNA and a pair of Outside primers were used to PCR amplify the genomic region of interest. Two mLs of Outside PCR products were used for a second PCR reaction using Inside PCR primers. One of the Inside primers is biotinylated and allowed for the isolation of and primer extension on a PSQ HS 96 pyrosequencer (Qiagen) according to the manufacturer’s instructions, and CpG sites were quantified, from 0% to 100% methylation, using the Pyro Q-CpG software.”

  1. RESULTS

Human Embryonic Kidney (HEK) 293: why not neuronal or glial cells from the start.

We started with 293HEK cells since they can be easily transfected. Many neuronal cells are difficult to transfect into due to their slower growth rate and DNA uptake. We had also tested the neuroblastoma SH-SY5Y but achieved only 22% transfection efficiency compared to 86% in 293HEK cells.

What is the percentage of transfection.

We transfected a similarly sized plasmid fragment (T7-BGH amplicon from pcDNA3.1(+) of 197 bp) into 12 replicates of 293HEK and used the same fragment and primers to generate a standard curve. By performing qPCR of gDNA derived from these transfected cells and normalized against single copy genomic DNA, we calculated the transfection efficiency at 86%.

2.2 Fig 2A compared to fig 2B: CpG-1 results in 2A compared to pyrogram 2B look confusing. Percentages are not comparable eg control 2A is 7% control 2B pyrogram is 6%.  Please check. Or leave pyrogram out because pyrogram is one analysis, fig 2A represents multiple analyses with SEM.

We apologize for the confusion. We had wanted to add actual raw pyrogram data besides bar graphs generated from a graphing software. Each “bar” in 2A represents the mean and SEM of methylation values for four replicates in each group. The pyrogram in 2B represents data for one typical sample. We edited the legend to indicate that the pyrogram is for one sample.

Mention the number of replicates.

All of the experiments were performed with four replicates (N=4 per group), not 3 as previously indicated in the first version of the manuscript. We have added the number of replicates in the figure legends.

Are replicates from different passages.

For each experiment, we seeded wells of samples using the same passage cells. We used the same passage to preclude the possibility of passage-dependent methylation differences confounding the study.

Why using SEM instead of SD?

We tend to use SEM instead of SD for cell line data since they tend to show only little variability across samples in a group. We typically use SD for clinical studies involving a large cohort size with greater variability. For MAOA, we presented results using both SD in the text for ANOVA and SEM in the graphs (to be consistent with other figures) due to the larger variability observed among samples within each group.

Mention statistical test that is used.

We used a Student’s t-test and ANOVA. We added the statistical test into a short methods section and legends.

2.3 Only the media were changed ? Thus no tranfection during the 4 weeks period took place? Just to be sure. Can you mention the percentage of dying cells? Please clarify my questions.

For the persistence experiment, no additional transfections took place. However, cells were split 1/5 whenever they became confluent and passaged further for 4 weeks. The reviewer is astute in asking about the percentage of dying cells, since transfection reagents can often lead to cytotoxicity and cell death. We had initially used Lipofectamine2000, which led to excessive (>50%) death of 293 cells. During our optimization process, we switched to X-tremeGENE 360 and used less DNA and transfection reagent, since the amount of transfected DNA reflected 100% probe DNA and no non-probe sequence such as those found on plasmids. 293 and AtT-20 cells are relatively robust in terms of withstanding the cytotoxicity associated with various transfections reagents. There was no noticeable cell death or “floaters” after switching to X-tremeGENE 360. Also, all groups, including negative controls, were exposed to the transfection reagent to control for cell toxicity and death. If there were any dead cells, they were washed off by PBS prior to cell collection so that they would not interfere with downstream molecular analysis.

2.5 ‘We also tested mouse pituitary cells to determine whether the findings in 293HEK can be recapitulated in a different cell-type and species.’ Reviewer appreciate the different cell-type, however why not using  the human neuronal or glial cells. You are interested in humans not in mice , right? Suggestion reviewer: show results in neuronal or glial cells.

We wanted to validate the DNA probe in a mouse line with the purpose of using the same probes in vivo using a mouse model. We agree with the reviewer that another important cell line would be a neuronal or glial cell line. We initially tested the SH-SY5Y cell line but found that it had relatively lower transfection efficiency compared to the 293HEK cell line. We have purchased SK-N-MC cells to repeat the experiment in this neuroblastoma cell line. However, it will take several months before we are able to complete similar experiments. We have added the need to test neuronal cells as a limitation.

  1. METHODS.

Mention X-tremeGENE 360 Transfection Reagent instead of X-tremeGENE 360

Done. Thank you.

Brief description of  conditions for bisulfite pyrosequencing Is appreciated.  The remark ‘Conditions for bisulfite pyrosequencing have been published elsewhere ‘ is not sufficient.

We have now added a brief description of the procedures for both bisulfite pyrosequencing and qRT-PCR.

Which primer design program did the authors use?

For both mouse and human FKBP5, these were designed and optimized by the corresponding author prior to the advent of software for designing bisulfite PCR primers.